# The Usefulness of the Regression-Based Normed SKT Short Cognitive Performance Test in Detecting Cognitive Impairment in a Community Sample

**DOI:** 10.3390/diagnostics14192199

**Published:** 2024-10-02

**Authors:** Mark Stemmler, Melina Arnold, Katya Numbers, Nicole A. Kochan, Perminder S. Sachdev, Henry Brodaty

**Affiliations:** 1Department of Psychology, Friedrich-Alexander University (FAU) Erlangen-Nuremberg, 91052 Erlangen, Germany; melina.arnold@fau.de; 2Centre for Healthy Brain Ageing (CHeBA), Discipline of Psychiatry and Mental Health, School of Clinical Medicine, University of New South Wales, Sydney 2052, Australia; k.numbers@unsw.edu.au (K.N.); n.kochan@unsw.edu.au (N.A.K.); p.sachdev@unsw.edu.au (P.S.S.); h.brodaty@unsw.edu.au (H.B.); 3Neuropsychiatric Institute, Prince of Wales Hospital, Sydney 2031, Australia

**Keywords:** cognitive assessment, regressions-based norming, dementia, MCI, sensitivity, specificity, Sydney Memory and Aging Study (MAS), short cognitive performance test (SKT)

## Abstract

**Background:** The SKT is a short cognitive performance test designed to assess impairments in memory and cognitive abilities such as attention and speed of information processing. In 2019, new regression-based norms for the English version of the SKT were calculated. This study has two aims: to establish valid cut-offs for distinguishing between no cognitive impairment, mild cognitive impairment (MCI), and dementia (1) and to cross-validate the new norms for detecting MCI and dementia in a community sample of older adults with clinical diagnoses (2). **Methods:** The validation sample included 143 older adults (mean age = 87.7, SD = 3.55) from the Sydney Memory and Aging Study (MAS Study). Participants were classified as having normal cognition, MCI, or dementia solely based on a consensus diagnosis; in addition, three tests (SKT, Mini-Mental State Examination (MMSE), and Addenbrooke’s Cognitive Examination III (ACE-III)) to measure cognitive impairment were applied. Sensitivity and specificity for all three tests, as well as bivariate correlations, were calculated. **Results:** The sensitivity of the SKT for the differentiation of cognitive impairment (MCI or dementia) from normal cognition was 80.6%. The convergence between the SKT and the consensus diagnoses was 70.3% for all three diagnostic groups. All correlations between the three tests and the consensus diagnosis were significant (*p* < 0.01). **Conclusions:** In sum, it can be stated that the SKT is a valid tool for detecting early stages of cognitive impairment, performing very well in discriminating between no cognitive impairment and cognitive impairment (MCI or dementia).

## 1. Introduction

In 2015, nearly 50 million people around the world were affected by various forms of dementia [1]. According to the GBD report, this number is projected to increase by 10 million each year. These figures highlight the global significance of dementia as a major social issue. Implementing preventive and therapeutic measures in the early stages of the disease can slow down its progression and delay the need for long-term care [2,3]. Currently, there is no therapy available to cure dementia. However, there are therapies designed to slow down or alleviate the symptoms of the illness. Drugs based on monoclonal antibodies, in particular, offer hope for Alzheimer’s disease specifically. The controversial effects of early monoclonal antibody drugs in this class, like Solanezumab or Aducanumab, on cognitive decline were hotly debated [4], and studies have shown that both drugs do not slow cognitive decline [5]. In contrast, the two drugs, Lecanemab and Donanemab, introduced sometime later, have shown promising results when it comes to slowing cognitive and functional decline in people diagnosed early with Alzheimer’s disease [6,7]. However, it remains unclear how effective the new drugs are compared to possible side effects [7].

Early detection of dementia during the prodromal stage, when targeted interventions are most effective, is crucial. In order to achieve this goal, it is essential to have reliable tests that can detect cognitive decline as soon as possible. Various studies have shown that established tests, such as the Addenbrookes’s Cognitive Examination (ACE-III [8]) and the Mini-Mental Status Examination (MMSE [9]), are valid tools for detecting cognitive impairment in older adults [10,11]. However, both tests have several weaknesses. One weakness is that there is only one form of the test. Therefore, if the test is administered more than once, it is possible that performance will be biased by recall of the last assessment (practice effect [12]). There is a lack of studies that focus on practice effects when testing for MCI or dementia using tests that measure cognitive impairment. Another notable drawback is that the standardization for both tests does not take into account different variables that could influence performance on the test (e.g., education, age, ethnicity). To our knowledge, there are approaches for norms that include these factors (e.g., [10,13]), but so far, they are only results from different studies and not general norms. Therefore, the Short Cognitive Performance Test (SKT, German abbreviation for Syndrom-Kurz-Test) [14,15] could possibly be an addition to the established cognitive testing instruments.

The SKT is a brief cognitive assessment for the diagnosis of cognitive impairment, comprising nine subtests, with a total application time of no more than 15 min. Among these subtests, three (subtests II, VIII, and IX) measure memory, while the remaining six (subtests I and IV to VII) evaluate attention. Based on the extracted number of factors, additional factors such as executive function or word fluency have been identified [16]. By examining the validity and reliability of the SKT, many studies have confirmed its usefulness [17]. In addition, the SKT has been translated into several languages, like Spanish and Norwegian [18,19].

The SKT offers several advantages compared to other brief cognitive assessments. Firstly, there are five parallel test forms, enabling the test to be repeated to monitor the progression of cognitive decline over time without practice effects [14]. Another notable advantage of the SKT is that it takes into account a number of factors in its scoring. Several studies (e.g., [20]) have shown that performance on cognitive tests depends on demographic factors and that these factors may influence cut-off points for norming. It is therefore important to include factors such as age, intelligence, and gender in the norms. Since 2015, an updated norming process has enhanced the usability of the SKT.

A detailed description of the new norming procedure and its results (regression-based norms) can be found in the SKT manual [14]. For the new norming procedure, multiple regression analyses were used to predict expected SKT raw scores for each subtest based on age, sex, intelligence, and their interactions in a large sample of older healthy persons. Based on the magnitude of the difference between observed and expected scores, either 0, 1, or 2 deviation points were given for each subtest, with more points indicating lower performance than expected. The resulting total scores range from 0 to 18. Higher points indicate lower-than-expected cognitive performance. Hence, the new norming approach embraced a new regression-based methodology [21]. Traditional norming procedures standardize an individual’s raw score, which is compared to typical scores from the reference group [22]. If, for instance, age is the norming predictor, an individual’s raw score is compared to scores of same or similar-aged people. Such traditional norms of subgroups are fine for categorial norming predictors such as school type but become somewhat arbitrary for continuous variables like age. For a traditional standardization, test developers divide the continuous norming predictor into arbitrary intervals and compute standardized scores within each subgroup using linear (e.g., T or IQ values) or area transformations (e.g., percentiles of norms). However, traditional norming methods may be biased if the same raw score produces a different normed score solely because of the person’s age group, especially if the individual’s age is near the threshold time point. Of course, the biases are strongly influenced by the width of the (age) subgroups [22]. Regression-based norming of the SKT, therefore, has an advantage in that there are no categorized age or intelligence groups; rather, this statistical norming approach means it is possible to use the exact age (in years) or the exact IQ. This precise norming reduces the biases mentioned above. For more comprehensive information on the English norming sample used for the SKT, please refer to the English manual [14].

The updated norming procedure for the SKT was initially validated using a German-speaking sample, yielding satisfactory outcomes [23]. It showed that, as suggested, there are differences in norms between German and English speakers. Predictors such as age or IQ varied in the strength of their influence on test results. Based on these findings, it can be said that it is important to have language-specific norms in order to achieve high diagnostic accuracy. Thus, these new regression-based norms based on data from the English norming sample [23] are a first step towards using the SKT in English-speaking countries and achieving accurate results.

To our knowledge, there are no valid cut-offs to differentiate between the diagnostic groups of no cognitive impairment, Mild Cognitive Impairment (MCI), and dementia for the English SKT. Further, this new norming approach has not yet been applied to an English-speaking sample. The aim of this paper is, therefore, twofold: (1) to establish valid cut-offs for distinguishing between no cognitive impairment, MCI, and dementia, and (2) to assess both the criterion and concurrent validity of the SKT in English-speaking older adults.

## 2. Materials and Methods

### 2.1. Participants

The proposed study is a validation study. For this purpose, data from the Sydney Memory and Aging Study (MAS Study) [24] were used. The MAS is a longitudinal study. However, based on the aim to validate the measurement quality of the SKT for each participant, only the values from one measurement point (wave 6 or wave 7) were used. The study, therefore, has a cross-sectional design. Participants for the original MAS were aged 70–90 years old and were recruited from two electoral districts in the Eastern Suburbs of Sydney, Australia, between 2005 and 2007. From 8914 individuals approached, 1037 were selected for the baseline sample. Inclusion criteria at baseline were the ability to read and write English sufficiently well to undergo psychometric evaluations and fill out self-report questionnaires. Individuals were excluded if they had psychiatric conditions, acute psychotic episodes, or had a current or previous diagnosis of multiple sclerosis, motor neuron disease, developmental disabilities, aggressive cancers, or dementia. Participants were further excluded if they scored below 24 on the MMSE at baseline. Participants gave written consent to partake in this research, which was approved by the University of New South Wales Human Ethics Review Committee (HC 05037, 09382, 14327, 190962).

Every two years, MAS participants undertook a detailed assessment with a trained research assistant (called a ‘wave’), and they completed a comprehensive neuropsychological test battery, medical history, medical exam, and a series of questionnaires. Clinical diagnoses were made at each wave by an expert consensus panel that considered all available neuropsychological, clinical, and informant-reported data [24]. At each wave, MCI was diagnosed using international consensus criteria [25], and dementia was diagnosed according to the Diagnostic and Statistical Manual of Mental Disorders-IV [26]. All of the patients who were diagnosed with dementia according to the DSM-IV also met the criteria for a major neurocognitive disorder according to the DSM-V [27]. For the current study, participant data from waves 6 and 7 (approx. 12–14-year follow-ups) were considered, as wave 6 was the first time the SKT was administered in MAS. Of the 317 participants who completed waves 6 and 7 of MAS, a subset of 140 completed the SKT from February 2018 to February 2020. In order to limit the burden on the elderly and to keep the testing process economical, the MAS team decided to include only a limited number of tests. That is why other established tests, such as the MOCA, are not included in these tests. The administration of the SKT at waves 6 and 7 is also the reason why the participants are relatively old.

### 2.2. Cognitive Measures

#### 2.2.1. SKT

The SKT is a short cognitive performance test assessing impairments in memory and attention. The test is scored on an 18-point scale, with lower scores indicating better cognitive performance, and is adjusted for age, sex, years of education, and IQ [14]. For the present validation study, age, sex, and years of education were obtained through MAS questionnaires and interviews, where IQ was conceptualized as crystalized and fluid intelligence. Crystalized intelligence was measured using the F-A-S Test, a verbal ability test that is strongly influenced by school education. The F-A-S is a subtest of the Neurosensory Center Comprehensive Examination for Aphasia (NCCEA [28]), and fluid intelligence was measured using the Digit Symbol Coding (DSC) subtest of the Wechsler Adult Intelligence Scale [29]. For norming and reasons of comparisons, results from both intelligence tests were transformed into Wechsler value points, with a mean (M) of 10 and a standard deviation (SD) of 3 [30].

To ensure comparability to the original norming sample, the results for the current sample were also categorized into three groups according to the traffic-light coding method developed by Stemmler and colleagues [14]: no cognitive impairment (green), MCI (yellow), and suspected dementia (red). Eighty-nine people (63%) were classified as green (no cognitive impairment), forty-nine people (35%) were classified as yellow (MCI), and two people (1.4%) were classified as red (suspected dementia).

#### 2.2.2. MMSE

The MMSE [9] is a well-established test of global cognitive function used to screen for dementia. The MMSE is an 11-question measure that tests different areas of cognitive function, including orientation, attention, recall, and language. It uses a 30-point scale, where a lower score indicates a higher level of cognitive impairment. Generally, a score of 24 or below is considered indicative of cognitive impairment (possible dementia), distinguishing it from normal cognitive function [9].

#### 2.2.3. ACE-III

The ACE-III [8] is a well-validated brief assessment of cognitive functioning that takes 12 to 20 min to complete. The ACE-III consists of five subscales: attention/orientation, memory fluency, language, and visuospatial functioning, with a maximum total score of 100. Lower total scores indicate poorer cognitive functioning, and a score of 88 or below is recommended as a possible indicator of dementia [31].

#### 2.2.4. Data Collection

The SKT was administered to participants at either wave 6 or wave 7. One patient had SKT data from both waves and was excluded from the study as the objective was to conduct a preliminary cross-sectional analysis. The MMSE was administered to participants at both wave 6 and wave 7. For those participants who were administered the SKT at wave 6, the wave 6 MMSE score was used for calculations. For those patients who were administered the SKT at wave 7, the wave 7 MMSE score was used for calculations. The ACE-III was administered to all participants at wave 6. Wave 6 ACE-III scores were also used for participants who completed the SKT at wave 7 (N = 106). To calculate the sensitivity and specificity of each test, test data were matched to the consensus diagnosis at the corresponding wave, excluding participants who had a consensus diagnosis but no test result (SKT, MMSE, ACE-III) or the reverse. For the calculation of correlations between tests and consensus diagnosis, missing data were excluded pairwise; see Table 1 for an overview of the data collection timeline. In wave 6, ethics approval to administer the SKT is received near the end of the wave, resulting in a low N. In wave 7, participants with previous diagnoses of dementia who could only handle limited neuropsychological testing did not receive the SKT or the majority of the protocol. If a participant did not provide informant data about their activities of daily living (ADL) or instrumental activities of daily living (IADL), the team responsible for making the consensus diagnosis would not have felt confident in making a nuanced diagnosis of mild cognitive impairment (MCI) versus normal cognition (NC). Similarly, if a participant had a lot of missing neuropsychological or other medical history data, the panel would not have felt confident in making a diagnosis. All assessments conducted after March 2020 were conducted over the phone due to COVID-19, and as a result, the SKT could not be administered to approximately one-third of the wave 7 participants. 

#### 2.2.5. Consensus Diagnoses

At each wave, participants were brought to a consensus review meeting where a panel of neuropsychiatrists, psychogeriatricians, and neuropsychologists discussed all available clinical, neuropsychological, laboratory, and informant-reported data to reach a consensus diagnosis. The consensus diagnosis did not involve the SKT. A range of cut-offs indicated when cases should receive consensus review, including impaired performance on neuropsychological tests (at least 1.5 SDs below published normative data on two or more domains). To diagnose MCI, the consensus criteria defined by Windblad et al. [22] were used. According to these guidelines, a person was diagnosed with MCI if they reported subjective cognitive complaints and showed cognitive impairment (SD ≤ −1.5) on one cognitive test if they did not have a diagnosis of dementia, and if they had no or minimal impairment in instrumental activities of daily living (for more detailed and additional information see [24]). A diagnosis of dementia was based on the Diagnostic and Statistical Manual of Mental Disorders, Fourth Edition (DSM-IV [26]) criteria, which include the cognitive impairment (SD ≤ −1.5) on at least two cognitive tests, with one test being memory specific, that is sufficiently severe as to cause impairment in daily functioning (e.g., Bayer Activities of Daily Living score ≥ 3.0). Participants who had complete neuropsychological test data and did not meet the criteria for an MCI or dementia diagnosis were classified as ‘normal cognition’ at each wave. For both waves, 111 consensus diagnoses were made. For the present study, eight participants were excluded because they had a non-amnestic form of MCI (nMCI). This decision was based on the finding that people with nMCI do not have significant memory impairments [32], which is what the SKT measures. Finally, seventy people (69%) were classified as green (no cognitive impairment), 15 people (14%) were classified as yellow (MCI), and 16 people (16%) were classified as red (suspected dementia). Figure 1 shows this distribution of the SKT classification.

#### 2.2.6. Comparing Group Status across Measures

Figure 1 presents a line diagram with the colors of the SKT traffic-light system as the categories, the SKT raw scores as the category axis, and the frequencies of the sum scores as the y-axis. Based on these values, we chose 3 and 11 as cut-off values for the traffic light system. That is, participants with a score from 0 to 3 were classified as green, indicating no cognitive impairment; participants with a score from 4 to 10 were classified as yellow, indicating MCI; and participants with a score of 11+ were classified as red, indicating suspected dementia. For the ACE-III, individuals with scores higher than 88 were categorized as not cognitively impaired, while individuals with scores lower than 83 were categorized as possible dementia. Those with scores between 83 and 88 inclusive were classified as possible MCI [20]. Regarding the MMSE, participants with scores above 26 were categorized as having no cognitive impairment, those with scores of 26 and 25 as MCI, and those with scores lower than 25 as possible dementia [9].

Finally, diagnostic classifications from each measure were compared with the participant’s consensus diagnosis, which was based on all available data and considered to be the gold standard.

### 2.3. Statistical Analysis

First, descriptive statistics were generated for the validation sample. Then, specificity and sensitivity values were calculated by cross-classifying the three levels of cognitive impairment (no cognitive impairment, MCI, dementia) with the different cognitive test results. To assess concurrent validity, bivariate correlations (Kendall’s Tau) were calculated between the consensus diagnoses and the classification of the SKT, the MMSE, and the ACE-III. To better understand the test results, all three tests were transformed into categorical variables (0 = no cognitive impairment, 1 = MCI, and 2 = dementia). Next, receiver operator characteristics (ROC) analyses were used to calculate the area under the curve (AUC) for the total SKT, MMSE, and ACE-III scores. The AUC is a measure of the overall classification accuracy of a test, where 1 is perfect accuracy. AUCs were calculated for all three tests. Additionally, for the SKT, Mann-Whitney-U-Tests were performed, where the independent variable was the SKT categorization group (based on the traffic light coding system), while the dependent variable was the consensus diagnosis. To account for multiple comparisons, we employed the Bonferroni-Holm correction.

All analyses were carried out using the software SPSS 29 or R-Software 4.4.2.

## 3. Results

### 3.1. Sample Characteristics

The sample consisted of 143 people aged between 83 and 98 (mean age = 87.7). A total of 90 of the sample were female, and the remaining 53 were male. Further characteristics of the sample are shown in Table 2.

### 3.2. Diagnostic Congruency

Table 3 presents the congruency between consensus diagnoses and the SKT diagnoses. The overall classification congruency was 71/101 (70.3%). The SKT demonstrated the highest hit rate for the category of no cognitive impairment (82.9%) and the lowest hit rate for the category of dementia (12.5%). Importantly, the hit rate for MCI was much higher at 73.3% compared to the hit rate for dementia. In the group of participants without cognitive impairment, none received a red label from the SKT (indicating possible dementia), but 17% were given a yellow label (indicating possible MCI). Among those with a consensus diagnosis of MCI, none was assigned a red SKT label, though two participants received a green SKT label, suggesting no impairment.

### 3.3. Sensitivity and Specificity

Table 4 presents sensitivity and specificity values for the SKT, MMSE, and ACE-III. For all three tests, the consensus diagnosis is taken as the “true diagnosis” and is used to calculate sensitivity and specificity. Across each test, the sensitivity for distinguishing between no cognitive impairment vs. cognitive impairment (i.e., normal cognition vs. MCI or dementia) is the highest, while the sensitivity for distinguishing between normal cognition vs. MCI is lower. In addition, of the three tests, the SKT has the highest sensitivity in both categories. Regarding specificity, all three tests demonstrate relatively high values across both categories. The lower specificity is observed in the category of no cognitive impairment versus cognitive impairment for all tests. The ROC analysis was calculated using the cut-offs, which were determined on the basis of Figure 1.

Figure 2 shows the results for two comparisons for all three tests. The ROC curves confirm the results from Table 4 regarding sensitivity and specificity. The best results were obtained for the no cognitive impairment vs. cognitive impairment comparison. However, the curves for the SKT and the ACE-III are very similar, while the curve for the MMSE is worse, indicating low sensitivity compared to high specificity. These patterns are found across both categories.

Finally, Mann-Whitney U tests were performed to analyze the extent to which both categories (no cognitive impairment vs. cognitive impairment, normal cognition vs. MCI) differed significantly in their consensus diagnosis. The results of the tests correspond to the sensitivities and specificities. The bar plots for the corresponding tests are shown in Figure 3a,b. For the category “no cognitive impairment vs. cognitive impairment”, the result was significant (*U* = 384.00; *p* < 0.001). The result suggests that there is a significant difference in the SKT scores between individuals with no cognitive impairment and those with cognitive impairment.

For the category “normal cognition vs. MCI”, the result was also significant (*U* = 239.0; *p* < 0.001). Again, these results indicate a notable difference in SKT scores between individuals with no cognitive impairment and those with MCI.

### 3.4. Concurrent Validity

In Table 5, correlations between the dementia tests and the consensus diagnoses, according to Kendall’s Tau, are presented. All correlations, both between the psychometric tests and the consensus diagnoses, as well as between the psychometric test scores themselves, were significant. Furthermore, these results indicate that of all three tests, the SKT has the highest correlation with the consensus diagnosis.

## 4. Discussion

After the establishment of the new English norms for the SKT in 2021, we aimed to determine the optimal cut-off points for the SKT in an English validation sample that would distinguish between no cognitive impairment, MCI, and dementia in a sample of older adults. Based on a line graph, the optimal cut-off score for MCI was 3 and 10 for dementia. The line graph showed significant increases after the cut-offs but no comparable increases at other points. These results support the accuracy of the cut-offs and are consistent with the cut-offs found for the German version of the SKT [14]. The cut-off for MCI in the English version is one point lower than in the German version (3 instead of 4), but the cut-off for dementia is the same. For the English norming, we recommend a cut-off of 3, especially if the mean age of the sample is high (c. f [33]) (here it was about 88 years); otherwise, a cut-off of 4 should be considered. These results suggest that the basic structure of the SKT remains very similar, although research [23,34,35] has shown that the strength of influence of different factors (e.g., age, gender, intelligence) varies between different language-specific norms.

The second objective was to validate the new regression-based norms for the SKT in a sample of older English-speaking adults. Together, the results indicate that the SKT is a valid instrument for discriminating between older adults with no cognitive impairment and those with cognitive impairment (i.e., MCI or dementia). In this context, the cut-off value of 3 seems to be effective in distinguishing between the two groups. This is a very encouraging result, as the primary aim of the SKT is to detect cognitive impairment as early as possible. The main focus of the SKT is not on distinguishing between MCI and dementia but between no cognitive impairment and cognitive impairment. The SKT performs very well in this area and, indeed, was better at distinguishing between these two groups compared to the MMSE and ACE-III, two commonly used cognitive (screening) tests. We also decided to test the sensitivity between normal cognition and MCI because MCI is a valid predictor of developing dementia. However, MCI may be harder to detect than dementia because the cognitive impairment is not as severe as in dementia. Fortunately, the sensitivity for normal cognition vs. MCI was only slightly lower than the sensitivity for no cognitive impairment vs. cognitive impairment. These results suggest that the SKT is also able to detect milder forms of cognitive impairment. These findings underscore the importance of early use of dementia screening tests, such as the SKT, because earlier detection of pathological cognitive decline provides an opportunity to start interventions that slow cognitive decline. Additionally, the SKT exhibits high psychometric properties, effectively distinguishing between cognitive impairment and normal cognitive function. This ensures that cognitively healthy older adults do not place undue strain on the healthcare system with unnecessary treatments. Additionally, results revealed that the differences between mean SKT scores across the groups of normal cognition, MCI, and dementia were significant. This is an indication that, although the power is not overwhelming for some comparisons, there is a clear difference in SKT performance among people with different clinical diagnoses.

The English norming calculation assumes that factors such as age, IQ, and sex have varying influences on SKT performance across different populations. The evidence supports the notion that language-normed dementia screening tests are relevant. Compared with other tests, the SKT has excellent sensitivity. However, there are only small differences in sensitivity and specificity between the SKT and the ACE-III, a well-validated screening test for cognitive impairment. It is important to note, however, that the SKT is shorter to administer and easier to score than the ACE-III, which is an important benefit. The MMSE performed less well in this sample. The MMSE performs best in terms of specificity but performs poorly in terms of sensitivity, which is a serious drawback.

These findings are supported by the correlations between the dementia tests and the consensus diagnoses. The correlation between the SKT and the consensus diagnosis is the highest, consistent with the good sensitivity of the SKT. The high correlation between the consensus diagnosis and the SKT suggests that they measure the same construct, cognitive impairment. In addition, the correlation between the ACE-III and the SKT is at a medium level. The ACE-III also correlates with the consensus diagnosis, so it is evident that all three measure the same overall construct. These findings are consistent with the very good sensitivity of the SKT and the ACE-III. The fact that the correlation between the MMSE and the consensus diagnosis is the lowest and that the correlations between the MMSE and the other two dementia tests are relatively low is also consistent with the poor sensitivity of the MMSE. Thus, the correlations, in combination with the other results, support the hypothesis that the SKT and the ACE-III measure the construct of cognitive decline at a valid level.

### Limitations

Although results indicate good validity of the SKT, there are some limitations. The first limitation is that the norms used for ACE-III and MMSE did not account for age or education, so it is possible that both tests would have performed better if these corrections had been made. However, to our knowledge, there are only exploratory papers (e.g., [36,37,38] for specific samples with corrections for variables that could be associated with cognitive decline. As there is no manual with corrected norms available, we have decided to use the uncorrected norms that are commonly used by most applications of both tests. Another limitation of this study is that the ACE-III was administered only in wave 6 of the MAS, meaning that ACE-III test scores for 102 participants were obtained from wave 6 and compared to wave 7 MMSE, SKT, and consensus diagnoses for these participants. This could be problematic as there could be up to a 4-year gap between the two waves, during which time cognitive function may have changed. Therefore, it is possible that the sensitivity and validity of the ACE-III test may be underestimated in this paper. As the sample size would only consist of 33 individuals if we had used data from the same wave for the ACE-III, we acknowledged this limitation and suggested obtaining a larger sample. As stated earlier, the sample size was limited, and the participants were relatively old. Furthermore, the participants were drawn from two different waves of a 12 to 14-year study, which may have resulted in non-random attrition bias. This implies that the majority of individuals who performed the SKT and were included in the sample had favorable cognitive abilities and/or an increased cognitive reserve, having remained in the study for 14 years, some with a diagnosis of MCI or dementia. Therefore, it is probable that the sample was not wholly representative of older adults, especially at that age. Taking into account the relatively small sample size, the possible non-random attrition bias, and the relatively old age of the sample, it can be concluded that the present study provides enough evidence for the very good psychometric properties of the SKT. However, further studies with larger samples and a wider age range are needed to confirm the results. It is noteworthy that intelligence was assessed using the aggregated scores from two tests, which represent the fluid and crystallized components of intelligence. The Digit Symbol test is a clear measure of fluid intelligence [39]. The FAS tests verbal fluency, which is a measure of crystallized intelligence. However, the time-limited testing of the FAS also includes a fluid intelligence component [40]. Therefore, both tests can be considered as reliable estimators of IQ. A full-scale measurement was too time-consuming and burdensome for older adults. Of course, as part of the daily testing routine, the estimate of IQ can be based on educational years in the absence of a premorbid IQ value.

## 5. Conclusions

The results of the studies show that the SKT has very good psychometric properties and compares favorably with other established psychometric tests for dementia. The test is also quick and easy to administer, and different parallel test versions are available. This allows an individual’s performance to be measured over time and can help healthcare professionals identify early changes that may indicate early signs of dementia pathology. The combination of these factors points to the high potential of the SKT in testing for cognitive impairment. Due to the above limitations, further studies are needed to confirm the results of this study.

## Figures and Tables

**Figure 1 diagnostics-14-02199-f001:**
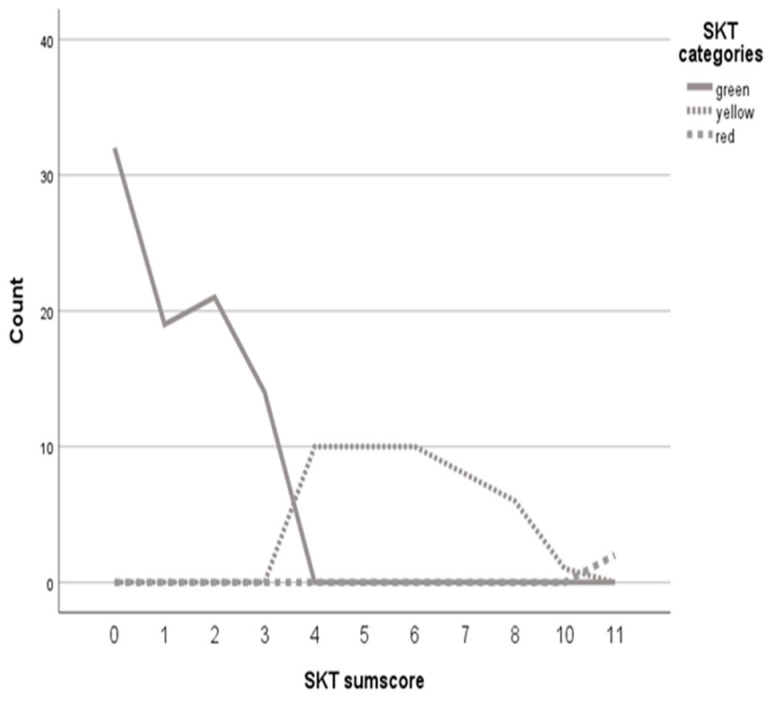
Frequencies of SKT raw scores based on the categorization of cognitive impairment.

**Figure 2 diagnostics-14-02199-f002:**
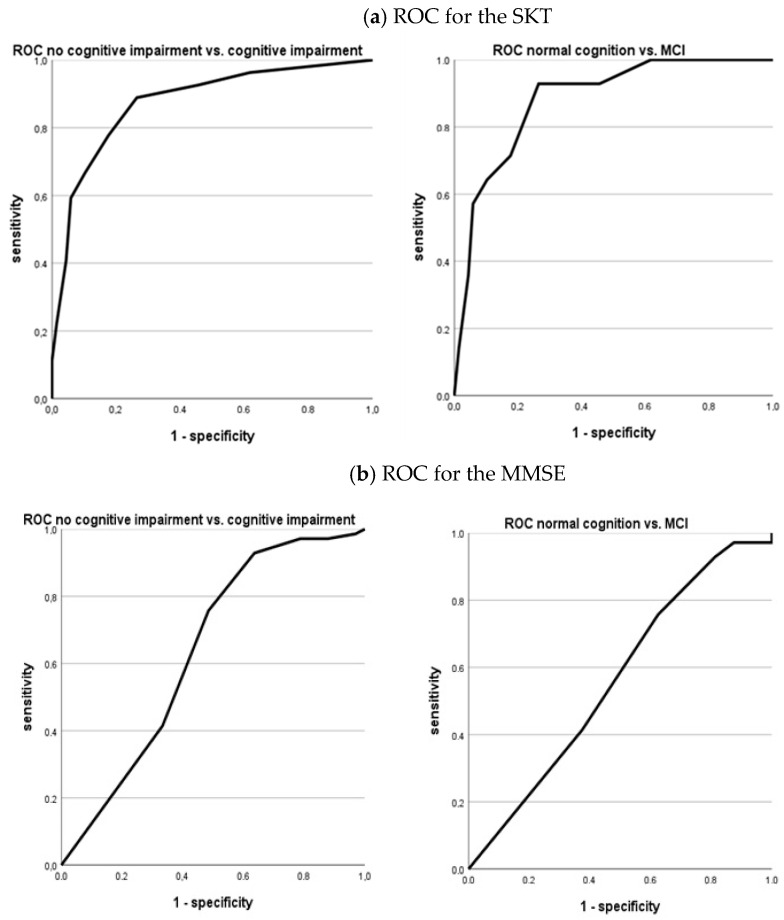
ROC curves for both categories of the three different tests.

**Figure 3 diagnostics-14-02199-f003:**
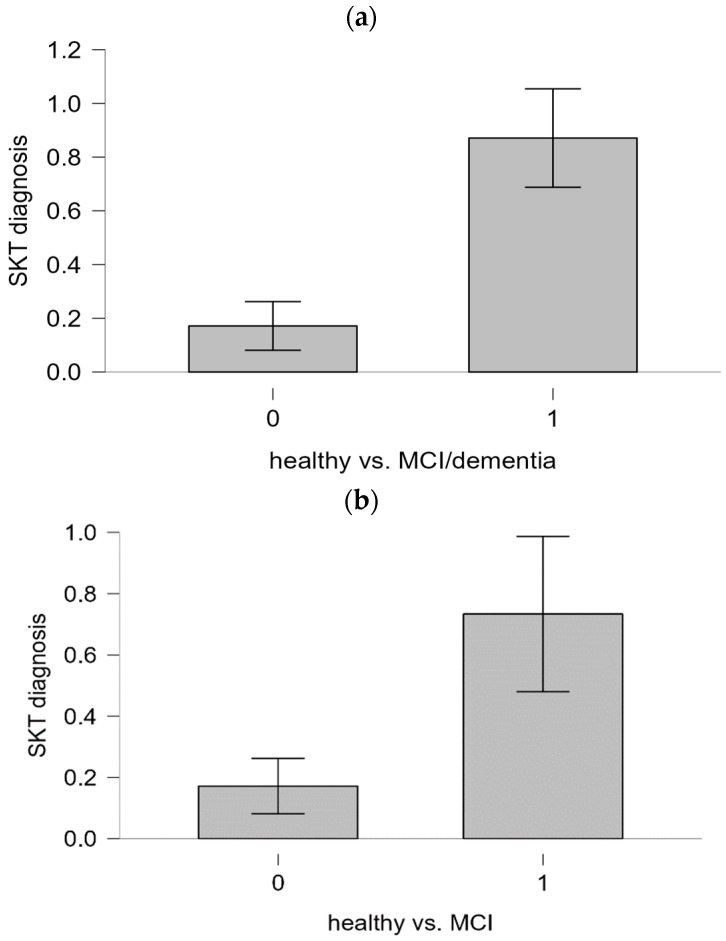
Mann-Whitney U test plots. (**a**) Mann-Whitney-U test no cognitive impairment (0) vs. cognitive impairment (1). (**b**) Mann-Whitney-U-test normal cognition (0) vs. MCI (1).

**Table 1 diagnostics-14-02199-t001:** Overview of the tests performed with the corresponding number of subjects (*n*; validation sample).

	Wave 6	Wave 7	Total Number of Persons Used for Calculations
Consensus Diagnosis	33	78	111
SKT	34	106	140
MMSE	142	109	143
ACE-III	142	0	142

**Table 2 diagnostics-14-02199-t002:** Descriptive statistics for validation study: age, Digital symbol Coding, F-A-S Test, SKT, ACE-III, and MMSE.

	N	Min	Max	Mean	SD
Age	143	83	98	87.73	3.54
Sex *	143	0	1	0.63	0.49
Years of Education	143	6	24	12.63	3.46
Digital Symbol Coding	137	12	78	42.14	12.22
F-A-S Test	139	12	71	37.78	12.86
SKT	139	0	11	2.99	2.84
ACE-III	142	63	100	89.58	5.96
MMSE	143	24	30	28.87	1.41

* 0 = male; 1 = female.

**Table 3 diagnostics-14-02199-t003:** Congruency between the consensus diagnosis and the SKT rating.

	SKT	Green (No Cognitive Impairment) N (%)	Yellow (MCI) N (%)	Red (Suspected Dementia) N (%)	
Consensus Diagnosis					
No cognitive impairment		**58 (82.9)**	12 (17.1)	0 (0)	70
MCI		4 (26.7)	**11 (73.3)**	0 (0)	15
Dementia		2 (12.5)	12 (75)	**2 (12.5)**	16
Total		64	35	2	101

Note: percentages apply to horizontal marginal cell totals. MCI = mild cognitive impairment. The values for which there is an agreement between the SKT and the consensus diagnosis are shown in bold type.

**Table 4 diagnostics-14-02199-t004:** Sensitivity and specificity for different comparisons for SKT, MMSE, and ACE-III.

		Sensitivity	Specificity	Area under the Curve
SKT	no cognitive impairment vs. cognitive impairment	80.6	82.9	87.5
	normal cognition vs. MCI	73.3	82.9	88
MMSE	no cognitive impairment vs. cognitive impairment	21.2	97.1	63.7
	normal cognition vs. MCI	12.5	98.6	56.3
ACE-III	no cognitive impairment vs. cognitive impairment	70	84.1	83
	normal cognition vs. MCI	72.7	86.6	83.7

Note. The consensus diagnosis is taken as the “true diagnosis” and is used to calculate sensitivity and specificity for all three tests.

**Table 5 diagnostics-14-02199-t005:** Correlation between the dementia tests and the consensus diagnosis, according to Kendall’s Tau.

	Consensus Diagnosis	SKT	MMSE
SKT	0.60 **		
MMSE	0.32 **	−0.24 **	
ACE-III	0.54 **	−0.32 **	0.34 **

Note. ** *p* < 0.01. Higher scores in the SKT represent more cognitive impairment.

## Data Availability

Dataset available on request from the authors.

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
