# Peer review of "The Usefulness of the Regression-Based Normed SKT Short Cognitive Performance Test in Detecting Cognitive Impairment in a Community Sample"

_diagnostics, 2024, doi:10.3390/diagnostics14192199_

Round 1
Reviewer 1 Report
Comments and Suggestions for Authors
Dear Authors,
Thank you for your submission on the evaluation of the SKT test for cognitive impairment. Your study addresses a critical topic in dementia research, and your methodological approach is sound in many respects. However, I recommend the following revisions:
1. Abstract: -Missing critical information about the methodology (e.g., participant details, testing conditions, and statistical analyses) which should be briefly summarized in the abstract.
-Limited detail on the statistical significance of the results. Include a brief mention of the statistical methods used (e.g., regression analysis) and the significance of results (e.g., p-values). (Lines: 15-33)
2. Introduction
-A more explicit problem statement is needed, focusing on what previous studies missed or why current tools are insufficient. Strengthen the gap analysis by specifying what previous studies failed to address and why this particular study is necessary.
Lines: 35-114
3. Methods
- The statistical methods used for data analysis are not well explained. The text lacks specific details. Provide a more thorough explanation of the statistical analysis performed (e.g., regression assumptions, handling of multicollinearity, outliers).
- Missing information on how missing data (e.g., participants excluded from waves) was addressed. Clarify how missing data was handled during analysis and justify exclusions.
Lines: 119-187
4. Results
- The results section lacks clarity in presenting inferential statistics and does not provide confidence intervals or p-values. Include p-values, confidence intervals, and effect sizes to provide a more robust interpretation of the findings
- The figures and tables could be better integrated into the text for clearer interpretation. Improve the description of the statistical analyses (e.g., regression outputs) and discuss the significance of the relationships.
Lines: 188-277
5. Discussio
- Your discussion does not adequately address the limitations of the study, particularly regarding sample size, selection biases, and generalizability.
- You overgeneralize some findings without sufficient backing from the data presented. I would suggest to avoid overgeneralizing the results, and provide more nuance in interpreting the clinical utility of the SKT. Thank you very much
Author Response
Reviewer 1
Abstract: -Missing critical information about the methodology (e.g., participant details, testing conditions, and statistical analyses) which should be briefly summarized in the abstract.
-Limited detail on the statistical significance of the results. Include a brief mention of the statistical methods used (e.g., regression analysis) and the significance of results (e.g., p-values).
We have worked on improving the abstract and tried to include all details as suggested. We are sorry, due to the 250 word-limit, it wasn't possible to include all of your points.
-Introduction: A more explicit problem statement is needed, focusing on what previous studies missed or why current tools are insufficient. Strengthen the gap analysis by specifying what previous studies failed to address and why this particular study is necessary.
Thank you very much. We added a section about the weaknesses of the most often-used cognitive screening tests and how the SKT could be a possible alternative, in order to solve the mentioned weaknesses.
-Methods: The statistical methods used for data analysis are not well explained. The text lacks specific details. Provide a more thorough explanation of the statistical analysis performed (e.g., regression assumptions, handling of multicollinearity, outliers).
We are sorry, but in the present study, no regressions were performed; instead, our article was based on the already completed new regression-based norms. As mentioned above, detailed information on this methodology can be found in Stemmler, M., Lehfeld, H. and Erzigkeit, A. (2019). In addition, we have added a sentence stating that the results of the other norming can also be found in the English manual, in order to make it clearer that the norms already existed and that our paper only wanted to prove their validity (see lines 108 - 109)
-Missing information on how missing data (e.g., participants excluded from waves) was addressed. Clarify how missing data was handled during analysis and justify exclusions.
We have added a paragraph about the handling of missing data in our analyses (see line 206 – 207). In addition, we wrote a short description of why we excluded the person with the SKT score at both waves. Exclusion criteria for the entire MAS can be found in the section on Inclusion Criteria (line 137).
-The results section lacks clarity in presenting inferential statistics and does not provide confidence intervals or p-values. Include p-values, confidence intervals, and effect sizes to provide a more robust interpretation of the findings.
We apologize again for the misunderstanding. As mentioned above, we didn't calculate regression-based norms, we just use the new regression-based norms that have been calculated and published in the manual (Stemmler, M., Lehfeld, H. and Erzigkeit. A. (2019)). This is also the reason why there are no statistical values for the regressions. For regressions and Mann-Whitney U tests, significant results are highlighted.
- The figures and tables could be better integrated into the text for clearer interpretation. Improve the description of the statistical analyses (e.g., regression outputs) and discuss the significance of the relationships.
We tried to integrate the text for clearer interpretation. As mentioned before, there were no regression analysis and therefore no regression outputs.
Discussion: -Your discussion does not adequately address the limitations of the study, particularly regarding sample size, selection biases, and generalizability. -You overgeneralize some findings without sufficient backing from the data presented. I would suggest to avoid overgeneralizing the results, and provide more nuance in interpreting the clinical utility of the SKT.
We agree with the limitations of the study. In our revision we have tried to make the limitations clearer and added a sentence about the importance of further research; we also used more nuance in interpreting the results (see lines 420 – 424 and 440).
Reviewer 2 Report
Comments and Suggestions for Authors
Dear Authors,
Thank you for the valuable and applied study.
Please remove the line 111-118 of the last paragraph of the introduction. The description of the participants in this study should be written in the method section.
Please write the type of study in the first line of the method. Draw a schematic diagram of the sample selection method and specify how to exclude the samples at each stage in the study.
The MMSE is a screening test and the Addenbrookes’s Cognitive Examination (ACE-III) is a diagnostic test. Please specify for what purpose the SKT test is used? Screening or diagnosis, which one?
The Addenbrookes’s Cognitive Examination (ACE-III) has high sensitivity and specificity for educated people not for all people. Please mention the reason why MOCA test was not used. Also Give the reason for choosing the age of 70 to 90 years according to the reference.
Because it was expected that the age of 65 years and above was selected, and they were divided into three categories: young elderly (65-75), middle-aged elderly (75-85) and elderly/old elderly (more than 85 years old) then statistical analysis based on The same classification was done.
Best regards,
Author Response
Reviewer 2:
-Please remove the line 111-118 of the last paragraph of the introduction. The description of the participants in this study should be written in the method section.
In these lines, there is the description of the sample used to compute the new regression-based normings. The aim of our paper is to investigate the quality and validity of these new normings. Therefore, we used the sample described in the Participants section. However, a small part of the sample in our paper was also part of the sample for the norming procedure. For the sake of clarity and transparency, we have therefore briefly described the norming sample in the theory section.
-Please write the type of study in the first line of the method. Draw a schematic diagram of the sample selection method and specify how to exclude the samples at each stage in the study.
We have added the study type in line 134. The schematic is a very nice idea. However, we have used the data from two waves of the MAS. As mentioned in line 222, information on the selection process could be found in the paper by Sachdev et al. (2010).
-The MMSE is a screening test and the Addenbrookes’s Cognitive Examination (ACE-III) is a diagnostic test. Please specify for what purpose the SKT test is used? Screening or diagnosis, which one?
The SKT is a small testing battery used to diagnose cognitive impairment. We have added this information to line 69.
-The Addenbrookes’s Cognitive Examination (ACE-III) has high sensitivity and specificity for educated people not for all people. Please mention the reason why MOCA test was not used. Also Give the reason for choosing the age of 70 to 90 years according to the reference.
In order to limit the stress on the elderly and to keep the testing process economical, the MAS team decided to include only a limited number of tests. The MOCA was not included in these tests. As the SKT was only administered at wave 6 and 7, we have no influence on the selection of cognitive tests. The administration of the SKT at wave 6 and 7 is also the reason why the participants are relatively old. We have added this information see lines 158 - 161.
- Because it was expected that the age of 65 years and above was selected, and they were divided into three categories: young elderly (65-75), middle-aged elderly (75-85) and elderly/old elderly (more than 85 years old) then statistical analysis based on The same classification was done.
Due to the fact, explained above, that we used data from the longitudinal MAS, in which the SKT was only assessed in waves 6 and 7, when all participants were relatively old, the age range was not chosen. In addition, because of the regression-based norms that included age, we didn't divide the participants into groups. We apologize for the possible misunderstanding, but there was no categorization
Round 2
Reviewer 1 Report
Comments and Suggestions for Authors
Dear authors,I would like to express my gratitude for your diligent efforts in addressing the feedback provided. Your responses to my queries were clear and comprehensive, and I highly appreciate the attention to detail in clarifying and expanding on the points I raised. The revisions you have made have been carefully considered and are in line with the suggestions given.
Kind regards
Author Response
Reviewer 1: Dear authors, I would like to express my gratitude for your diligent efforts in addressing the feedback provided. Your responses to my queries were clear and comprehensive, and I highly appreciate the attention to detail in clarifying and expanding on the points I raised. The revisions you have made have been carefully considered and are in line with the suggestions given. Kind regards
Thank you very much!
Reviewer 2 Report
Comments and Suggestions for Authors
Dear authors,
Thanks for your responses.
However Some conflicts are sill remained. For example, the meaning of the type of study is that you specify whether your study is cross-sectional, analytical, comparative or instrumental. You have stated in line 134 that this is an original study of ageing memory. In terms of research methodology, the type of study is not introduced in this way.
Despite your concern about clarifying the status of the participants, there is no need to introduce the participants even partially in the introduction. Please delete this section.
Please highlight all the changes to make it easier to compare the new version with the old version.
Rock curves related to the comparison of tools is one of the most important parts related to the statistical analysis of this study. Please include these graphs in the findings section and provide explanations for each of the graphs.
Best regards,
Author Response
Thank you very much for the comments of reviewer 2. The comments helped us to improve the manuscript, even better. I comment on the reviewer’s remarks by leading you through their concerns.
Reviewer 2:
-However, some conflicts still remained. For example, the meaning of the type of study is that you specify whether your study is cross-sectional, analytical, comparative or instrumental. You have stated in line 134 that this is an original study of ageing memory. In terms of research methodology, the type of study is not introduced in this way.
We have tried to make the type of the study clearer by adding a short description of the section 2.1 participants (see line 160).
-Despite your concern about clarifying the status of the participants, there is no need to introduce the participants even partially in the introduction. Please delete this section.
We deleted this section.
Rock curves related to the comparison of tools is one of the most important parts related to the statistical analysis of this study. Please include these graphs in the findings section and provide explanations for each of the graphs.
As the reviewer suggests, we moved the ROC curves from the appendix to a figure in the paper. We also added an interpretation of the results of the ROC curves.
Please highlight all the changes to make it easier to compare the new version with the old version.
Therefore, we resubmit the manuscript with track changes.
Again, we think that the manuscript has much improved, thanks to the expert reviewers.
